

# An Internally Consistent Dataset of $\delta^{13}$C-DIC in the North Atlantic Ocean - NAC13v1

Meike Becker[1,2], Nils Andersen[3], Helmut Erlenkeuser[3], Matthew.P. Humphreys[4], Toste Tanhua[1], and Arne Körtzinger[1,2]

[1]GEOMAR, Helmholtz Center for Ocean Research, Kiel, Germany
[2]Christian Albrechts University Kiel, Kiel, Germany
[3]Leibniz-Laboratory for Radiometric Dating and Isotope Research, Christian Albrecht University, Kiel, Germany
[4]Ocean and Earth Science, University of Southampton, Southampton, UK

*Correspondence to:* Meike Becker (mbecker@geomar.de)

**Abstract.**

The stable carbon isotope composition of dissolved inorganic carbon ($\delta^{13}$C-DIC) can be used to quantify fluxes within the carbon system. For example, knowing the $\delta^{13}$C signature of the inorganic carbon pool can help to describe the exchange between ocean and atmosphere as well as the amount

of anthropogenic carbon in the water column. The measurements can also be used for evaluating modeled carbon fluxes, for making basin wide estimates, studying seasonal and interannual variability or decadal trends in interior ocean biogeochemistry. For all these purposes, it is not only important to have a sufficient amount of data, but these data must also be internally consistent and of high quality.

In this study, we present a $\delta^{13}$C-DIC dataset for the North Atlantic, which has undergone secondary quality control. The data originate from oceanographic research cruises between 1981 and 2012. During a primary quality control step based on simple range tests obviously bad data were flagged. In a second quality control step, biases between measurements from different cruises were quantified through a crossover analysis using nearby data of the respective cruises and absolute val-

ues of biased cruises were adjusted in the data product. the crossover analysis was possible for 22 of the 29 cruises in our dataset and adjustments were applied to 10 of these. The internal accuracy of this dataset is 0.017‰.

The dataset is available via CDIAC at http://cdiac.ornl.gov/oceans/ndp_096/NAC13v1.html, doi:10.3334/CDIAC/OTG.NAC13v1.

# 1   Introduction

Stable carbon isotope ratios are utilized as a tracer in several applications in marine carbon research. Particularly the stable carbon isotope ratio of dissolved inorganic carbon ($\delta^{13}$C-DIC) can be used to enhance the understanding of carbon related processes ranging widely from the estimation of



glacial circulation changes (Curry and Oppo, 2005) to testing the performance of ecosystem models

(Schmittner et al., 2013). By observing the temporal development of the lightening of the inorganic carbon pool due to the uptake of $CO_2$ originating from the burning of $^{13}$C-depleted fossil fuel carbon, a phenomenon also known as oceanic $^{13}$C Suess effect, an estimation of the anthropogenic carbon fraction of DIC is possible (Gruber et al., 2002; Körtzinger et al., 2003; Olsen et al., 2006, 2010; Quay et al., 2007; Racapé et al., 2013). Furthermore, $\delta^{13}$C can provide information concerning the

quantification of biological processes such as net community production (Quay et al., 2009). Using the stable carbon isotope signature facilitates the distinction between anthropogenic, biological and physical drivers of the carbon system.

A sample's stable carbon isotope ratio, $\delta^{13}$C-DIC, is expressed as per mil deviation from that of the commonly used standard material Vienna Pee-Dee Belemnite (V-PDB)(Coplen, 1995).

$$\delta^{13}\mathrm{C} = \left( \frac{^{13}R}{^{13}R_{\mathrm{PDB}}} - 1 \right) \cdot 10^3 \qquad (1)$$

with $^{13}R$ being the ratio of the two stable carbon isotopes $^{13}$C and $^{12}$C in the sample.

For basin-wide carbon flux estimates, studies of seasonal variations, or interannual trends it is important to have a dataset of sufficiently high coverage both in space and time. Moreover, the dataset should be free of systematic differences between measurements carried out by different laborato-

ries and on different cruises. However, both criteria are not easily met. Since Isotope Ratio Mass Spectroscopy (IRMS), the common method to analyze $\delta^{13}$C-DIC data, is a very time consuming and expensive technique that cannot be performed at sea, data coverage has remained relatively poor. Therefore, several efforts have been made to assemble a dataset containing as many cruises as possible.

For oceanic $\delta^{13}$C-DIC data this has been done first by Kroopnick (1985) who provided an analysis of the distribution of $\delta^{13}$C-DIC in the world's oceans. Over the years more data was accumulated and different data collections emerged (Gruber et al., 1999; Quay et al., 2003, 2007; Schmittner et al., 2013). During recent years, databases like GLODAP (Global Ocean Data Analysis Project) and CARINA (Carbon dioxide in the Atlantic Ocean) were created for carbon-related parameters

(Olsen et al., 2016). These projects did not only assemble the data but also conducted a secondary quality control so that systematic biases between individual cruises could be identified and adjusted for (Tanhua et al., 2009; Velo et al., 2009; Tanhua et al., 2010a; Pierrot et al., 2010). Relative to other parameters such as total alkalinity or DIC, however, the dataset for $\delta^{13}$C-DIC is still small and disorganized. Therefore, no secondary quality control in which deep water samples from different

cruises at the nearby locations, so called crossovers, are compared to each other could be carried out within these collections. Several new cruises have become available for the North Atlantic so that now the present crossover study could be performed for this area. This crossover analysis features



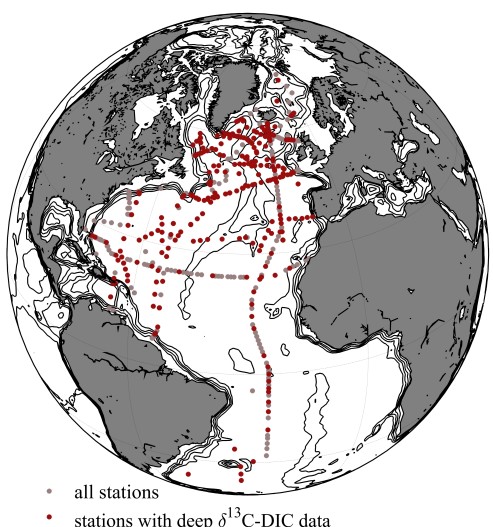

· all stations
· stations with deep $\delta^{13}$C-DIC data

**Figure 1.** Map of all stations with $\delta^{13}$C-DIC data used in this dataset. Data from deeper than 1500 m was available only for the stations in dark red, so only these stations were used for the crossover analysis.

29 cruises of which 22 could be compared quantitatively. Cruises without a quantitatively evaluable crossover were qualitatively related to the corrected dataset.

Please note, that for applying the crossover inversion routine we assume that the deep water masses (below 1500 m) are only to a negligible amount influenced by changes due to an increasing amount of anthropogenic carbon. Since the detected differences between some cruises were not consistent with a slowly increasing amount of anthropogenic carbon we think that this consistent dataset is a important step for improving the study of carbon isotope dynamics in the upper 1500 m. In regions

for which also the deeper water masses have been shown to contain a high amount of anthropogenic carbon, were neglected crossovers with cruises that took place long before or long after the respective cruise. We believe that no temporal trends have been removed, or created by the 2nd QC procedures employed here. However, care should be exercised for calculating $C_{ant}$ accumulation in water below 1500 m.




**Table 1.** Information about sample handling and measurements for those cruises where the $\delta^{13}$C data have not been published elsewhere.

| Cruise ID | Expocode | Laboratory | Analysis period | Sample handling | PI |
|---|---|---|---|---|---|
| 1 | 06MT19941012 | [1] | 9/2002 - 12/2002 | 200 $\mu$L HgCl$_{2\text{sat}}$/ 100 mL sample | A. Körtzinger / H. Erlenkeuser |
| 2 | 06MT1997-M39 | [1] | 1/1998 - 2/2000 | 200 $\mu$L HgCl$_{2\text{sat}}$/ 100 mL sample | A. Körtzinger / H. Erlenkeuser |
| 3 | 06MT1999-M45 | [1] | 7/2000 - 6/2002 | 50 $\mu$L HgCl$_{2\text{sat}}$/ 100 mL sample | A. Körtzinger / H. Erlenkeuser |
| 4 | 06MT20010507 | [1] | 12/2001 - 9/2002 | 50 $\mu$L HgCl$_{2\text{sat}}$/ 100 mL sample | A. Körtzinger / H. Erlenkeuser |
| 5 | 06MT20030723 | [1] | 3/2004 - 10/2004 | 100 $\mu$L HgCl$_{2\text{sat}}$/ 100 mL sample | A. Körtzinger / H. Erlenkeuser |
| 6 | 06MT20040311 | [1] | 1/2005 - 10/2005 | 200 $\mu$L HgCl$_{2\text{sat}}$/ 100 mL sample | D.W.R. Wallace / H. Erlenkeuser |

[1] Leibniz Laboratory for Radiometric Dating and Isotope Research, Kiel, Germany

## 2  Data Provenance and Structure

This dataset comprises data and metadata from 29 research cruises/campaigns from several international research groups, in total 6068 samples. Some of these consist of multiple cruises and one is a time series. For the crossover analysis, some consecutive cruises whose data were analyzed together were treated as one cruise. While the focus is on the North Atlantic, four cruises were included that also have stations in the Nordic Seas, and one cruise extends into the South Atlantic. Thereby, con-

sistency with extended quality controlled datasets for these regions is ensured. Since only deep (> 1500 m) samples of each cruise are compared in this study, only cruises with at least one deep station could be included in this analysis.

Figure 1 shows the locations of all stations with $\delta^{13}$C-DIC data that are part of this compilation. For cruises that have not been published elsewhere, Table 1 shows a summary of the respective

sample handling, the periods during which the samples were analyzed and the responsible PI. Some cruises had $\delta^{13}$C-DIC measurements over the entire depth range at every station, whereas others just had one or two stations with deep $\delta^{13}$C-DIC data. Most of the cruises were conducted in the subpolar North Atlantic, while the tropical region has relative poor coverage. The temporal and latitudinal distributions of the data are displayed in Figure 2. The data was collected in the North

Atlantic between 1981 and 2012, with the majority falling between 1990 and 2005. Considering the





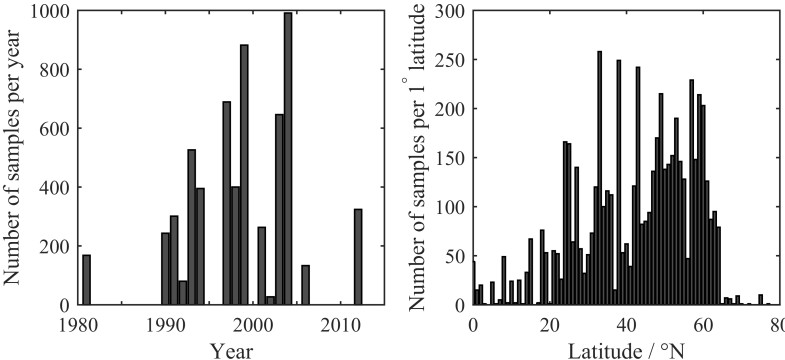

**Figure 2.** The temporal distribution of the presented dataset ordered by year (left panel) and the sum of all samples at each latitudinal degree (right panel).

seasonal distribution of the data, a bias towards summer time exists, especially towards late summer. The only two cruises which took place between January and March were located south of 42°N. The uncertainty of the $\delta^{13}$C-DIC samples analyzed by IRMS is usually reported to be between ±0.12‰ (Gruber et al., 1999) and ±0.03‰ (Quay et al., 2003).

The presented dataset consists of 20 columns of which the first 17 are cruise number, station, sampling number, day, month, year, latitude, longitude, maximal depth, maximal sampling depth, bottle number, cast number, temperature, salinity, depth, ctd salinity and pressure. Column 18 contains the adjusted $\delta^{13}$C-DIC data, column 19 a quality flag(C13f) and column 20 the QC-flag (C13qc, see Table 2). For bad data the quality flag was set to 'not measured' and therefore column 19 has only

two entries (2: good, 9: not measured). Cruises that could be quantitatively compared to each other by the 2nd QC have a '1' in the QC-flag. All others are flagged with '0'.

Additional parameters to most of the cruises can be found in either GLODAPv2 or CARINA. Only the most recent cruises 35TH20060521 and 74DI20120731 are not included in these datasets, but the individual cruise files can be found on CDIAC. The OMEX1NA data are only part of Carina.

The respective cruise numbers in GLODAPv2 and CARINA of the cruises shown in the NAC13v1-dataset can be found in the documentation.



**Table 2.** NAC13v1 data set parameter list, column names used in the data product and the respective units.

| parameter | data product parameter name | data product flag name | unit |
| --- | --- | --- | --- |
| NAC13v1 cruise number | cruiseno | | |
| Station | station | | |
| Sample number | nosamp | | |
| Day | day | | |
| Month | month | | |
| Year | year | | |
| Latitude | latitude | | °N |
| Longitude | longitude | | °E |
| Bottom depth | maxdepth | | m |
| Pressure of the deepest sample | maxsampdepth | | dbar |
| Bottle number | bottle | | |
| Cast number | cast | | |
| Temperature | temperature | | °C |
| Salinity | salinity | | |
| Depth | depth | | m |
| CTD salinity | ctdsal | | |
| Pressure | pressure | | dbar |
| $\delta^{13}$C-DIC | C13 | C13f, C13qc | ‰ |

## 3   Computational Analysis

In order to derive an internally consistent set of $\delta^{13}$C-DIC data in the North Atlantic all publicly available data in this area were assembled and quality controlled (QC) in two steps. At first, a primary QC was performed in order to identify obviously erroneous data, such as wrong positions, time stamps and depths. Also outliers were identified and then flagged by comparing the profiles of each cruise internally. After that, the secondary QC procedure was conducted employing a crossover analysis as described by Tanhua et al. (2010b). This MATLAB based software package compares two cruises at a time, searches for nearby stations, so-called 'crossovers', and calculates differences between all crossovers of the two cruises as additive offsets with the unit ‰. As criterion for identifying crossovers a maximum of 180 nm (3° of latitude) distance between stations was used. From these crossovers, the $\delta^{13}$C-DIC data collected deeper than 1500 m was compared on equal potential density. Based on the resulting offsets and standard deviations determined for each of these crossovers a suggestion for a possible adjustment was made. This suggestion was obtained by an





inversion routine using a Weighted Least-Square (WLSQ) and a Weighted Damped Least Square
(WDLSQ) model as described by Johnson et al. (2001). The cruise 33MW19930704-1 covers a long
distance and is assumed to have high quality data. Therefore this cruise was selected as core cruise
and weighted higher than the other cruises. Unfortunately this was the only cruise meeting these
two criteria. Several cruises from different years were in good agreement with the core cruise while
the other cruises were adjusted towards it. Choosing the appropriate distance criterion for crossover
locations is always a compromise between including as many statistically relevant crossovers as pos-
sible by selecting a large enough radius on the one hand and trying to have only crossovers between
stations that share similar oceanographic characteristics on the other hand. However, reducing the
crossover distance to 120 nm reduced the amount of crossovers and the number of cruises that could
be quantitatively compared to each other but did not significantly change the suggested magnitude
of adjustments of the remaining cruises. Therefore, the 3°x3° criterion was used instead. For some
crossovers in highly variable regions with deep water formation, such as the Labrador Sea and the
Nordic Seas, the standard deviation was decreased significantly by restricting the comparison depths
to >2000 m. Generally, offsets from crossovers in these highly variable regions, from cruises with
a relatively poor data precision or with just a few deep samples were considered in the model with
less influence, by weighting the offsets with their uncertainty. In Figure 3 all crossovers between the
cruises 06MT20030723 and 33MW19930704-1 are shown as an example, both for the uncorrected
as well as for the corrected dataset. All crossovers from the adjusted and the unadjusted dataset can
be found at: http://cdiac.ornl.gov/oceans/ndp_096/NAC13v1.html.

    Whether an adjustment was applied to the data was decided somewhat subjectively in each case
based on a combination of the shape and distribution of individual crossover differences and the
suggestions given by the inversion routine with knowledge about the sampling region. After applying
the adjustments, the inversion was conducted again and it was checked whether or not the adjustment
improved the overall consistency within the entire dataset. Temporal changes of the deep water
masses were only considered in this step of the routine when comparing the suggested corrections
and the corresponding crossover offsets between cruises in areas where also the deep water $\delta^{13}$C-
DIC was expected to change over time. In order to get a quantitative description of the internal
consistency of the final dataset, a weighted mean using the respective offsets of all crossovers and
their standard deviation was calculated (Tanhua et al., 2010a).

$$\mathrm{WM} = \frac{\sum\limits_{i=1}^{\mathrm{L}} \mathrm{D}(i)/(\sigma(i))^2}{\sum\limits_{i=1}^{\mathrm{L}} 1/(\sigma(i))^2} \tag{2}$$

    L refers to the total number of crossover, D to the respective offset of all crossover and $\sigma$ is their
standard deviation.

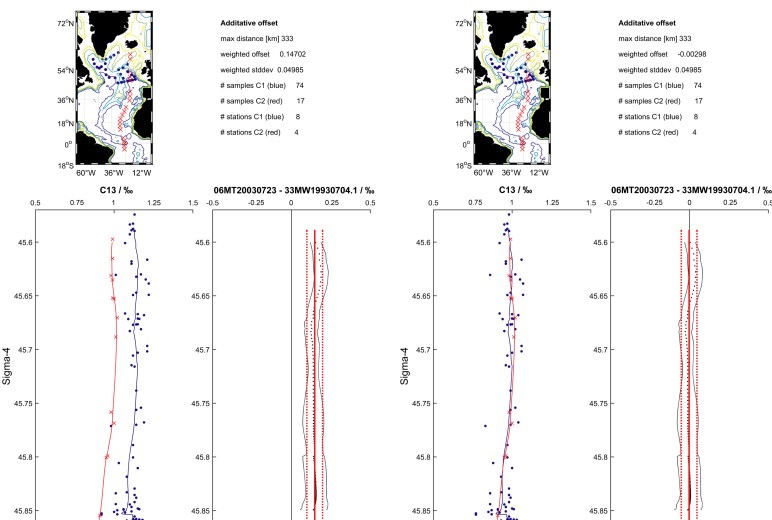

**Figure 3.** Crossovers between the cruises 06MT20030723 and the core cruise 33MW19930704-1. The left hand plot shows the original and the right hand plot the adjusted data. In both cases the distribution of the $\delta^{13}$C-DIC on equal density surfaces (left hand side) as well as the mean offset between both cruises (right hand side) is shown. The cruise 06MT20030723 was adjusted by -0.20 ‰.

## 4 Adjustments

The data of all cruises as well as locations are shown in Figure 4. The offsets, as well as the corrections suggested by the WDLSQ inversion routine, and the final adjustments are listed in Table 3. In Figure 5 the results of the WDLSQ inversion are shown before and after the adjustments were applied. Some cruises show quite big deviations from the core cruise. However, we do not know the reason for these biases. Besides the actual sample analysis in the laboratory, also different sampling routines on board the ship, insufficient poisoning and the sample storage time can cause these biases.

A detailed overview of the offset of each crossover in the original as well as the adjusted dataset is given in Table 4 in the supplementary information. Moreover, the evidence for our decision will be presented for each cruise.

### 4.1 06MT19941012, cruise #1

This cruise on the German R/V Meteor is also known as M30-2 (Körtzinger et al., 1998). The inversion suggested a correction of -0.07 ‰. The mean offset of all crossovers is 0.11 ‰ too high. Based on the crossover with the core cruise an adjustment of -0.07 ‰ was applied.





**Table 3.** Overview of all cruises in this dataset. The data of some cruises were combined for the analysis. For more information, please see the detailed description in the 'Adjustments' section. Both, the mean offsets and the corrections suggested by the WDLSQ inversion are shown for the original and the adjusted dataset. In the last column the applied adjustments are displayed. NC indicates that these cruises were not considered in the inversion since they had no statistically significant crossover and the core cruise is marked with C. Cruises with insufficient quality data are denoted 'poor' and not included in the further analysis.

| cruise ID | Expocode | Calculated offset | | Suggested adjustments | | Final adjustments |
|---|---|---|---|---|---|---|
| | | not adjusted | adjusted | WDLSQ | WDLSQ (adj) | |
| | | / ‰ | / ‰ | / ‰ | / ‰ | / ‰ |
| 1 | 06MT19941012 | 0.11 | -0.02 | -0.07(±0.10) | -0.01(±0.02) | -0.07 |
| 2 | 06MT1997-M39 | -0.02 | 0.02 | 0.01(±0.14) | 0.00(±0.01) | 0 |
| 3 | 06MT1999-M45 | 0.16 | -0.01 | -0.14(±0.09) | 0.00(±0.01) | -0.15 |
| 4 | 06MT20010507 | 0.16 | 0.00 | -0.24(±0.10) | 0.00(±0.01) | -0.30 |
| 5 | 06MT20030723 | 0.14 | 0.03 | -0.15(±0.09) | 0.00(±0.01) | -0.20 |
| 6 | 06MT20040311 | -0.14 | -0.02 | 0.10(±0.09) | 0.01(±0.01) | 0.10 |
| 7 | 316N19970717 | 0.17 | 0.02 | -0.06(±0.17) | -0.01(±0.01) | -0.05 |
| 8 | 316N19970815 | | | | | NC |
| 9 | 316N20030922 | | | | | NC |
| 10 | 316N20031023 | | | | | NC |
| 11 | 33RO19980123 | | | | | NC |
| 12 | 33MW19910711 | -0.02 | -0.02 | 0.00(±0.01) | 0.00(±0.01) | 0 |
| 13 | 33MW19930704-1 | -0.05 | 0.01 | 0.00(±0.01) | 0.00(±0.01) | C |
| 14 | 35TH20020611 | | | | | NC |
| 15 | 35TH20060521 | -0.39 | -0.02 | 0.24(±0.21) | -0.03(±0.05) | 0.25 |
| 16 | 58JH19920712 | | | | | NC |
| 17 | 58JH19940723 | | | | | NC |
| 18 | 64TR19900417 | | | | | poor |
| 19 | 74DI20120731 | -0.33 | -0.13 | 0.13(±0.28) | 0.12(±0.12) | 0 |
| 20 | OMEX1NA | -0.14 | -0.03 | 0.03(±0.13) | 0.02(±0.02) | 0 |
| 21 | 316N19810401 | -0.06 | 0.03 | -0.03(±0.10) | -0.01(±0.03) | 0 |



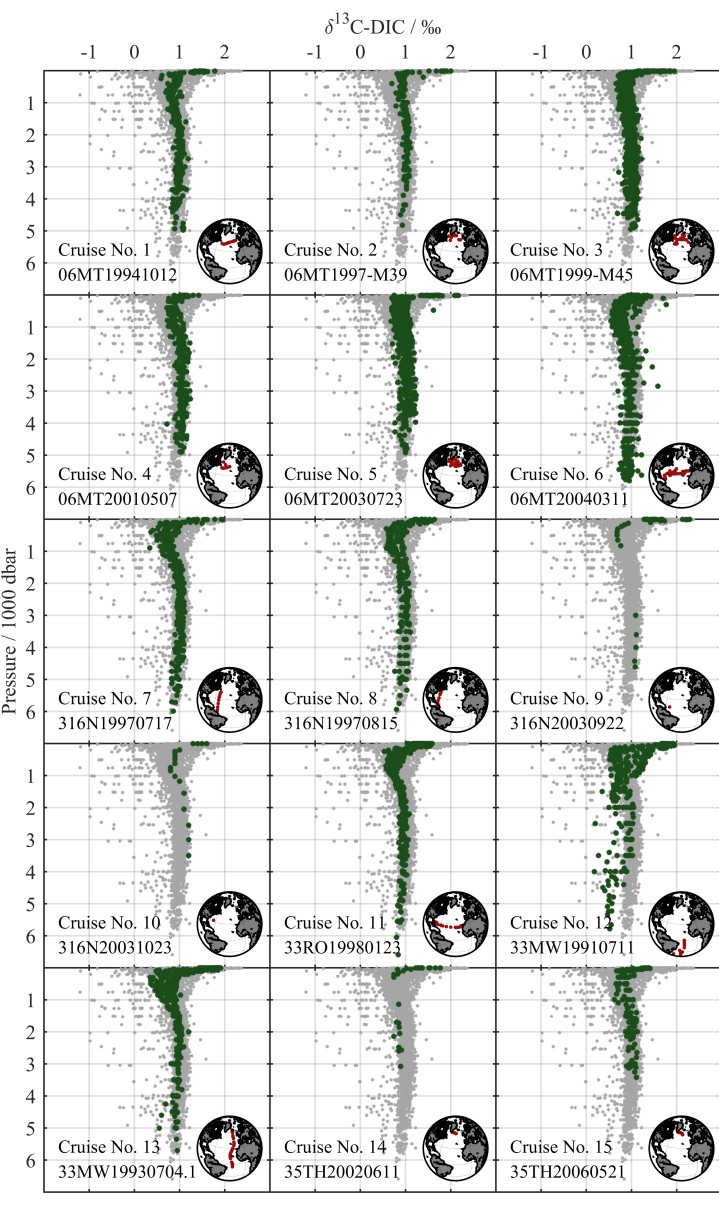

**Figure 4.** Adjusted $\delta^{13}$C-DIC profiles and locations of each cruise. The green profiles represent the data of the specific cruise whereas the gray dots show all profiles in the dataset. 





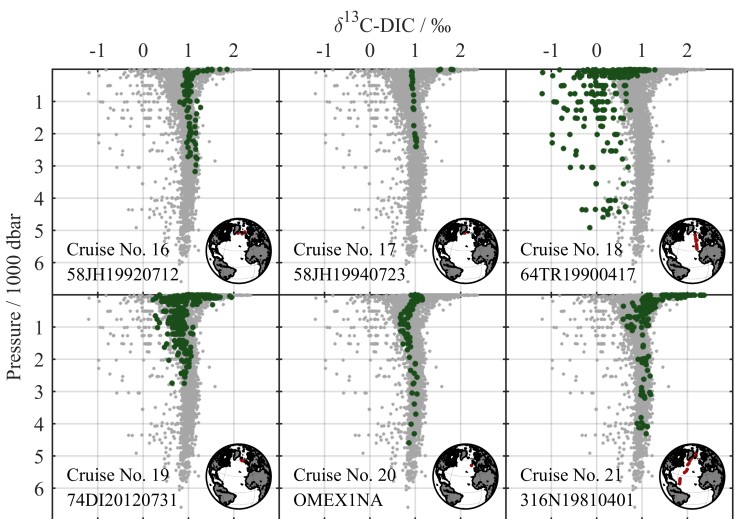

**Figure 4.**                                                                                    *continued from previous page.*

Adjusted $\delta^{13}$C-DIC profiles and locations of each cruise. The green profiles represent the data of the specific cruise whereas the gray dots show all profiles in the dataset.

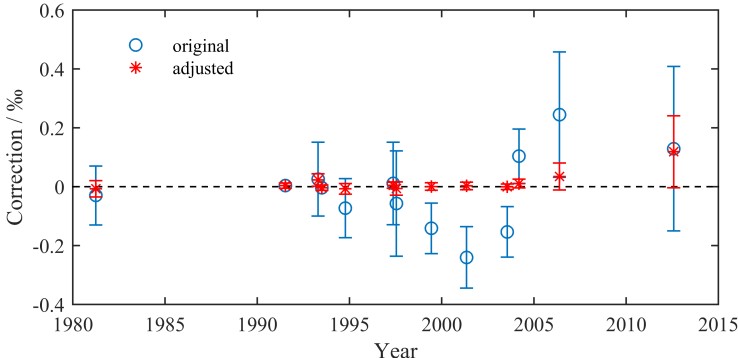

**Figure 5.** The results of the WDLSQ based inversion routine for the original (blue circles) and the adjusted dataset (red stars). The cruises are plotted at the time the data was collected vs. the suggested correction.

### 4.2    06MT19970515, 06MT19970707 and 6MT19970815, here referred to as 06MT1997-M39, cruise #2

These cruises are also known as M39 cruises with three legs of $\delta^{13}$C-DIC sampling (M39-2, M39-4, M39-5) (Körtzinger et al., 1999; Thomas and Ittekkot, 2001). Since each leg of this cruise had only





a few stations with $\delta^{13}$C-DIC samples, and all these samples were analyzed together, these cruises
were summarized for the crossover study. Both, the inversion routine and the single crossover with
the adjusted cruises show no evidence for an offset.

### 4.3    06MT19990711 and 06MT19990813, here referred to as 06MT1999-M45, cruise #3

These cruises are also known as M45-2 and M45-3 (Friis et al., 2005). Since both were analyzed
together, they were summarized for this crossover study. The inversion suggested a correction of -
0.15 ‰ and the mean offset of all crossovers was 0.16 ‰ too high. After applying this adjustment and
comparing this cruise to the adjusted dataset, the inversion routine still suggested a small correction.
Therefore, an adjustment of -0.20 ‰ was applied.

### 4.4    06MT20010507, cruise #4

This cruise is also known as M50-1 (Friis et al., 2007). The inversion routine suggested a correction
of -0.24 ‰, whereas the mean offset was 0.16 ‰ too high. Based on the southern crossover with
cruise 06MT20040311 and 316N19970717 an adjustment of -0.30 ‰ was applied.

### 4.5    06MT20030723, cruise #5

This cruise is also known as M59-2 (Friis et al., 2007). The correction suggested by the inversion
routine is -0.15 ‰ which matches with the positive offsets of the crossovers, except of those with
33TH20060521. Based on the crossover with the core cruise, an adjustment of -0.15 ‰ was applied.

### 4.6    06MT20040311, cruise #6

This cruise is also known as M60-5 (Tanhua et al., 2007). The inversion routine indicates that the
$\delta^{13}$C-DIC data of this cruise is 0.10 ‰ too low. Additionally, the mean offset shows that this data is
too low. An adjustment of +0.10 ‰ was applied.

### 4.7    316N19970717, cruise #7 and 316N19970815, cruise #8

These cruises followed the WOCE/GO-Ship standard lines A20 and A22 (Johnson et al., 2003). The
inversion suggests a correction of -0.06 ‰ for 316N19970717. It shows one crossover with cruise
06MT20040311 in which a significant positive offset is still visible after cruise 06MT20040311 was
corrected. Therefore, an adjustment of -0.05 ‰ was applied for cruise 316N19970717. The cruise
316N19970815 does not show a statistically significant crossover.

### 4.8    316N20030922, cruise #9, and 316N20031023, cruise #10

These cruises, that took place in the tropical western Atlantic, following the A20 and A22 lines, have
only one deep station each (Feely et al., 2008). The crossovers of these stations with both, the ad-





justed data of cruise 06MT20040311 and cruise 316N19970717 show a good agreement, suggesting
that no adjustment should be applied.

### 4.9    33RO19980123, cruise #11

This cruise (Lee et al., 2003) has one statistically insignificant crossover with the cruise 06MT20040311
and one with cruise 33MW19930704-1. Both seem to be in good agreement, suggesting that no ad-
justment should be applied.

### 4.10    33MW19910711, cruise #12, and 33MW19930704-1, cruise #13

The cruise 33MW19930704-1 was considered as core cruise in the present analysis (Forde et al.,
1996). The cruise 33MW19910711 extents into the south Atlantic and its crossover with cruise 13
shows no need for an adjustment.

### 4.11    35TH20020611, cruise #14, and 35TH20060521, cruise #15

The latter of these two cruises has a few quantitative crossovers, that show a high offset of -0.39‰.
Furthermore, the inversion suggests a correction of 0.24‰. The high variability of the sampling area
south of Iceland, as well as an increasing lightning of the deep water carbon pool over time don't
give an adequate explanation for this large deviation and, therefore, an adjustment of -0.25‰ was
applied. The cruise 35TH20020611 shows just a few qualitatively analyzable crossovers, that show
a lighter carbon pool compared to earlier cruises and a heavier one compared to the original data
of cruise 35TH20060521 (Racapé et al., 2013). After adjusting this cruise, both cruises, which were
analyzed in the same laboratory, are not in good agreement anymore which suggests that the earlier
cruise also has too low isotope values. However, in the absence of a statistically significant crossover
no adjustment was applied to this cruise.

### 4.12    58JH19920712, cruise #16, and 58JH19940723, cruise #17

These two cruises took place in a highly variable area (Gislefoss et al., 1995). No statistically relevant
crossover exists but the data are in good agreement with the core cruise and the other adjusted cruises
in that area.

### 4.13    64TR19900417, cruise #18

This cruise shows extreme scatter compared to all other cruises and, therefore, was not included into
the adjusted product (Rommets et al., 1991). When comparing crossover stations this cruise shows
a mean offset to other cruises of about -1.2‰.





### 4.14 74DI20120731, cruise #19

Both the inversion and the offset mean of the crossover suggest a correction of +0.13 ‰ for the cruise
(Humphreys et al., 2015). This most recent cruise took place near the Scotland-Iceland ridge where
the deep water masses cannot be assumed to be constant over time. All crossovers indicate a lower
$\delta^{13}$C-DIC of this cruise when comparing it with the others which is consistent with an increased
amount of anthropogenic carbon. Therefore, no adjustment was applied.

### 4.15 OMEX1NA, cruise #20

During the OMEX1 project in the North Atlantic $\delta^{13}$C-DIC samples were taken in January 1994
(Wollast and Chou, 2001). The data is in good agreement with the other cruises in this area and no
adjustment was applied.

### 4.16 316N19810401, cruise #21

The cruises 316N19810401, 316N19810416, 316N19810516, 316N19810619, 316N19810721,
316N19810821 and 316N19810923 are combined and usually named Transient Tracers in the Oceans
North Atlantic Study (TTO-NAS) (Brewer et al., 1986). The inversion does not suggest any correc-
tion for this dataset.

## 5 Conclusions

The finalized, quality controlled dataset of $\delta^{13}$C-DIC presented here consists of 22 cruises (some of
which consists of multiple legs) that have been quantitatively compared to each other and form an
internally consistent dataset. Seven cruises could not be quantitatively compared to the other cruises
due to a lack of crossovers and / or deep $\delta^{13}$C-DIC data. The internal consistency of the adjusted
dataset was calculated to be 0.017‰ based on Equation 2.

The database is available at CDIAC via http://cdiac.ornl.gov/oceans/ndp_096/NAC13v1.html,
doi:10.3334/CDIAC/OTG.NAC13v1.

## 6 Acknowledgment

We would like to thank all the people, both researchers as well as captains and crews, who spend time
at sea and in the lab collecting and measuring the samples and preparing the data that is presented
here and the PIs for sharing it. The measurements of the majority of previously not public $\delta^{13}$C data
was supported by the Deutsche Forschungsgemeinschaft (DFG) through SFB460 and this work was
funded by the Future Ocean Excellence Cluster Project CP1140. We also thank Aley Kozyr from
CDIAC for preparing the NAC13v1-website on CDIAC.



**Table 4.** This table shows an overview of all crossovers. The symbol ● in each row divides the table into a triangle in the upper right and one in the lower left of the table. In the upper right corner for each crossover the offsets of the offsets of the original dataset are listed. In the lower left corner the remaining offsets in the adjusted dataset are shown. Not statistically relevant crossovers are displayed with >, < and = indicating the tendency of not significant crossover. Please note that the offsets shown in the table result from $\delta^{13}C_{column} - \delta^{13}C_{row}$. All offsets are given in ‰.



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
