# Peer review of "An Internally Consistent Dataset of $\delta^{13}\text{C}$ -DIC in the North Atlantic Ocean - NAC13v1"

_Earth System Science Data, 2016_

## Referee Comment (RC1) · A. Olsen (Referee) · 30 Mar 2016

Review of "An Internally Consistent Dataset of $\delta$13C in the North Atlantic Ocean" by Becker et al.

This dataset is an important asset to ocean biogeochemistry, hopefully it may also inspire to global efforts. I have not problem recommending this for publication provided the following is dealt with.

Major issues

The consistency analysis relies on the use of crossovers and inversion. As Becker et al. states in the discussion paper, this is not always easy to do for 13C because of the limited spatial data coverage, and they had to increase the maximum crossover

distance to 3 degrees, compared to the 2 degrees, which is commonly used for DIC, alkalinity and nutrients, for instance. However, there certainly exists alternatives, several authors, including papers cited by Becker et al., have used Multilinear Regressions (MLR) to determine the ocean Suess effect. Such MLRs can also be used to evaluate data consistency, for example by developing MLRs using data from deeper than 1500 m. This is relatively easy to do, for example by developing an MLR based on the 33MW1993 core dataset, or based on all data in the set, and finding the biases when the MLR is used to determine 13C from each individual cruise. If the biases are of the same direction and magnitude as those determined with the crossover and inversion this would certainly add confidence to the adjustments. If satisfactory MLRs cannot be determined using data from >1500 m, I am certain that an MLR derived from data from the full water column would also reveal biases, when applied on data from deeper than 1500 m only.

This dataset is not very large, consisting of data from 29 cruises. A table listing all cruises, dates, PIs, and peer-reviewed citations for each would certainly be worthwhile and possible to include.

According to Table 1, some of the new data were analysed up to 8 years after the samples were collected, and some data sets were analysed over a period of approximately 2 years. There is a potential effect of storage on $\delta$13C samples, so it would very useful with some analyses of the effect of storage time on dataset accuracy, did you find any correlation between bias and time between collection and analysis in these data, or with scatter?

The collection miss the data from the Nordic Seas cruise 58GS20030922 (used in Olsen et al., 2006), these are available through CDIAC, for instance through GLODAPv2 cruise summary table, please include, these are probably the most extensive Nordic Seas 13C data available.

Issues with datafile

[Figure]

The data that were deemed bad are still available in the data file, but flagged 9. I suggest to remove them, this was done both in CARINA and GLODAPv2, and it is better to be on the safe side; flags are frequently ignored. Make sure the original data are available in the cruise data files at CDIAC.

In the datafile the "nosamp" and "cast" columns are empty. There is no point of including "nosamp" if there are no values. The "cast" column can be critical to any merging efforts, please make sure these numbers are included.

The "maxsampdepth" is largely empty, this is trivial to fill, please do so.

Cruise 13, station 83, maxdepth is -82, this cannot be right, please correct.

Minor issues

Table 3, Fig 4 & 5 has units after a backslash "/", please use parenthesis.

line 6, "making basin wide estimates".. of what, please specify

line 6, please include an "and" before "studying"

line 14, please delete "absolute", or revise, "absolute" can be understood in terms of by absolute magnitude (i.e. neglecting any negative sign)

line 15, Captial "T" in "the"

lines 24-30. I like this list of uses of 13C data. However, the abstract gives more, for instance 'help to describe the exchange between the ocean and the atmosphere', these should be mentioned in the main text as well, with citations to examples of these applications (I am curious about this example, and other readers may be so as well).

line 28, Olsen et al., 2010 did not use $\delta$13C data, but please feel free to include a citation to Olsen and Ninnemann, 2010 instead.

line 30-33, please include specific example for this application (citation is sufficient).

line 37, 'for basin-wide carbon flux estimates', please be more specific, what is meant,

air-sea fluxes? can this be done?

line 55, delete "the"

Fig 1., the data points are hard to see, please remove bathymetry.

line 60, please insert an "and" between "crossover" and "inversion"

line 64, well, I am sure that the dataset is also important for studying isotope dynamics below 1500 m, for example spatial variations should be present.

line 66, replace "were" with "we". Please also specify "long", how many years?

line 79-80. I do not understand this, what other extensively quality controlled C-13 data sets are there to ensure consistency with?

line 89, please replace "was" with "were"

line 92, please replace "which" with "that"

line 103, please include citations to GLODAPv2 (Olsen et al., 2016) and CARINA (Key et al., 2010)

line 104, please insert commas after "cruises" and "74DI20120731"

line 104, Carina in caps.

line 114, please specify which profiles were compared, 13C vs 13C or 13C vs other parameters? Please provide one or two examples of profiles with outliers. It would certainly be useful to include property-property figures in the primary QC step, for example AOU vs 13C.

line 116, Tanhua et al describes several types of crossover analyses, please specify which was used, e.g. 'running crossover'.

line 133, you may want to add that 120 nm was the commonly used distance in CARINA; PACIFICA, and GLODAPv2 so readers understand where this number comes

from.

line 136, the standard deviation .. of what, please specify.

Fig 3, please specify what the various vertical lines indicate, in the caption.

line 169. I do not completely understand, according to Table 4 in the supplement the crossover difference between the 06MT19941012 and the 33MW1993 cruise is not significant, still 0.07 is -indirectly-stated in the text. Please also clarify what it takes for a crossover to be significant.

line 193, "data" are plural, hence write "these data are" not "this data is", check all places were "data" is mentioned.

line 202, should be "these cruises, which", not "these cruises, that", please consult rules for which vs. that and also comma use ("which" should be preceded with comma, "that" shouldn't)

line 215, again, which vs. that.

lines 219-224. This passage is a bit confusing, please clarify. As I understand it, the 2002 Thalassa cruise data were not adjusted, but it had crossovers, why doesn't these data show up in Fig. 5?

line 227, this is not correct; the 58GS2003 cruise can be used and is available at CDIAC.

---

## Referee Comment (RC2) · A. Schmittner (Referee) · 5 Apr 2016

The authors present a secondary quality controlled dataset of d13C_DIC data from the North Atlantic. I compliment the authors for this and share reviewer Olsen's desire to see this extended to the globe.

In Schmittner et al. (2013) we used AOU versus d13C_DIC plots to find outliers and identify offsets (but this may not be described in detail in that paper). Anyway we did not correct the dataset for any offsets, but we noticed them and were concerned about it. So, I'm glad that here a systematic approach is used to identify and correct offsets. But I wonder if the AOU vs d13C_DIC approach may also be of value to use here. E.g. it is not clear to me that internal variability could not change d13C_DIC on interannual-decadal time scales even in the deep ocean. Due to the strong correlation to AOU any

change in the biologically sequestered carbon that affects d13C_DIC should be visible in AOU. Anyway, this is just a suggestion that the authors may want to consider for a revision.

(page 3) Anthropogenic d13C_DIC changes have been estimated by models (e.g. our 2013 paper mentioned above). These model results (available at http://people.oregonstate.edu/~schmita2/data/schmittner13bg/Model/) could potentially be used to estimate the effect in different regions.

(page 7) Why does cruise 33MW199930704-1 have high quality data? Were there objective criteria used to determine this?

(line 140) these cruises are 10 years apart and I could imagine that at high latitudes anthropogenic d13C_DIC could have an impact. (see comments above)

(Fig. 3) the font and figure is too small. It is not readable. Please increase size.

(line 182) "-0.20 permil" but in Tab. 3 -0.15 permil is listed. Please check this inconsistency.

---

## Author Comment (AC1) · 15 Jul 2016

**Response to Reviewer 1**

Thank you very much for this detailed review. It helped us improving the paper significantly. We hope, our answers consider all your comments and suggestions is a satisfactory way.

Please note, that changes we did in the manuscript or the dataset are italicized in the following and highlighted in red in the manuscript.

**Major issues**

- Certainly there exist other methods for estimating systematic differences within a

dataset. We decided for the crossover analysis since the Multi-parameter-MLR analysis is limited by the availability of other parameter such as AOU, DIC, $NO_3$. We now conducted a MLR based on the deep core cruise data. For cruises that are located in the North Atlantic, this MLR analysis reveals offsets in the same order and magnitude as the crossover inversion routine. For those cruises that reach into the Nordic Seas, the picture is more difficult due to the different water masses.

Answer:

*There is a paragraph added to the section 'Computational Analysis' that explains the MLR analysis:*

*Another method for revealing systematic deviations between different cruises is a regional multi-linear regression (MLR) (Wanninkhof2003, Jutterstrom2010). In this work, a MLR based on core cruise data (deeper than 1500 m) was used to verify the suggested corrections that resulted from the crossover analysis. Moreover, some cruises without a statistically evaluable crossover could now be related to the other cruises. The following equation was used,*

$$\delta^{13}C - DIC_{MLR} = -16.9 + 0.80 \cdot S - 0.080 \cdot \Theta - 0.0045 \cdot DIC \qquad (1)$$

*with $\delta^{13}C - DIC_{MLR}$ being the calculated $\delta^{13}C - DIC$, S the salinity, $\Theta$ the potential temperature in $^\circ C$ and DIC the DIC concentration in $\mu molkg^{-1}$. The DIC concentration was chosen because it is strongly related to changes in the isotope composition and DIC data were available for most cruises. Adding more parameters to the MLR, such as apparent oxygen utilization (AOU) or nutrient concentrations, did not improve the agreement between $\delta^{13}C - DIC$ and $\delta^{13}C - DIC_{MLR}$ of the core cruise and reduced the amount of cruises hat could be compared via the MLR analysis. The limitation of this method is, of course, that the further away in space and time the cruises are from the core cruise, the more likely an observed offset is real. Especially, the cruises reaching into the Nordic seas show*

*significant deviations, which are most likely real differences between the basins. Therefore, the offsets revealed by the MLR analysis were not taken into account for these cruises.*

*These sentences were added in the 'Adjustment' section:*

*For most cruises that took place in the North Atlantic, the offsets revealed by the MLR analysis were in the same order and magnitude as the suggested correction by the crossover inversion routine. Cruises reaching far into the Nordic Seas or the South Atlantic show huge differences, which are caused by different water mass properties in these areas.*

*Moreover, the results of the MLR analysis are also addressed in the detailed discussion of each cruise.*

- "This dataset is not very large, consisting of data from 29 cruises. A table listing all cruises, dates, PIs, and peer-reviewed citations for each would certainly be worthwhile and possible to include."
  Answer:
  *A table listing all cruises, dates, PIs and publications was added.*

- "According to Table 1, some of the new data were analysed up to 8 years after the samples were collected, and some data sets were analysed over a period of approximately 2 years. There is a potential effect of storage on $\delta^{13}$C samples, so it would very useful with some analysis of the effect of storage time on dataset accuracy, did you find any correlation between bias and time between collection and analysis in these data, or with scatter?"
  Answer:
  Yes, the new data were analyzed over a long period of time and also stored for a long time. I could not find any correlation between a cruises' bias or its scatter and storage time, analyzing period or volume of $HgCl_2$ added.
  *The following sentences were added to the Conclusion:*
  *The reason of the deviations between single cruises could not be revealed. There*

*was no correlation between a cruises' bias or its scatter and storage time, analyzing period or volume of HgCl$_2$ added.*

- "The collection miss the data from the Nordic Seas cruise 58GS20030922, these are available through CDIAC, for instance through GLODAPv2 cruise summary table, please include, these are probably the most extensive Nordic Seas 13C data available."
  Answer:
  We are aware of the cruise 58GS20030922. Its data were included into the analysis, but not into the final dataset, since this cruise had no crossover, not even a few samples within a crossover radius, with which it could be compared to the rest of the dataset. This was also the case for another cruise in the Nordic seas (74JC20120601). Now, we included both into the final dataset. However, since this dataset concentrates on the North Atlantic and some assumptions clearly don't hold for the Nordic Seas (3x3°, small anthropogenic influence on deep water masses) we suggest, that it would be a better choice to perform a consistency analysis focused on the Nordic seas alone, once there are enough cruises available. Also, the water mass properties of the core cruise, which was used for the MLR analysis, were too different from those of the Nordic Seas to reveal any reliable statement on systematic biases between these cruises.

  *We added three more cruises to the dataset (58GS20030922, 74JC20120601 and 74JC20140606), of which the first two are located in the Nordic seas and the latter one became recently available. Therefore, all figures, tables and also the absolute numbers of samples and cruises included in the presented dataset were updated in the paper.*

**Issues with the dataset**

- "The data that were deemed bad are still available in the data file, but flagged 9."
  Answer:

*The data flagged as bad were removed from the dataset.* All new original data has been submitted to CDIAC prior to submitting the paper.

- "In the datafile the "nosamp" and "cast" columns are empty. The "maxsampdepth" is largely empty, this is trivial to fill, please do so. Cruise 13, station 83, maxdepth is -82, this cannot be right, please correct."
  Answer:
  *The column 'nosamp' was excluded. The columns 'maxsampdepth' and 'cast' were correctly filled.*

**Minor issues**

All minor text issues were corrected. The manuscript was checked again for 'data were' and which vs. that.

- **Tab 3, Figure 2,3,4,5:** "has units after a backslash "/", please use parenthesis'"
  Answer:
  We prefer the backlash-version for axis labeling, which is standard in physical equations.

- **I 6:** ""making basin wide estimates".. of what, please specify"
  Answer:

- **II 24-30:** "I like this list of uses of 13C data. However, the abstract gives more, for instance 'help to describe the exchange between the ocean and the atmosphere', these should be mentioned in the main text as well, with citations to examples of these applications (I am curious about this example, and other readers may be so as well)."
  Answer:
  *We deleted 'help to describe the exchange between the ocean and the atmosphere'.*

However, measurements of surface ocean and atmospheric 13C with a high spatial and temporal resolution (for example on VOS) hold the potential to reveal seasonal as well as interannual changes in the isotope signature of air-sea gas exchange.

- **l 28:** "Olsen et al., 2010 did not use $\delta^{13}$C data, but please feel free to include a citation to Olsen and Ninnemann, 2010 instead."
*Changed to Olsen and Ninnemann (2010)*

- **ll 30-33:** "please include specific example for this application (citation is sufficient)."
Answer:
*Citation added (Gruber et al., 1998)*

- **l 37:** "'for basin-wide carbon flux estimates', please be more specific, what is meant, air-sea fluxes? can this be done?"
Answer:
*The sentence was changed to 'of carbon fluxes due to primary production'*

- **Fig 1:** "Fig 1., the data points are hard to see, please remove bathymetry."
Answer:
*Bathymetry was removed.*

- **l 64:** "well, I am sure that the dataset is also important for studying isotope dynamics below 1500 m, for example spatial variations should be present."
Answer:
For sure it is. But it is restricted by the basic assumption of a crossover analysis and we just wanted to be sure that it is handled with care in applications.

- **ll 79-80:** "I do not understand this, what other extensively quality controlled C-13 datasets are there to ensure consistency with?"

Answer:
We meant all the datasets that (hopefully) will come up in the future.

- **l 103:**"please include citations to GLODAPv2 (Olsen et al., 2016) and CARINA (Key et al., 2010)"
  Answer:
  *Both citations were included.*

- **l 114:** "please specify which profiles were compared, 13C vs 13C or 13C vs other parameters? Please provide one or two examples of profiles with outliers. It would certainly be useful to include property-property figures in the primary QC step, for example AOU vs 13C."
  Answer:
  For identifying outliers $\delta^{13}$C profiles were compared. We also made AOU vs $\delta^{13}$C plots now for those cruises with available AOU data. There were to outliers left.

- **l 116:** "Tanhua et al describes several types of crossover analyses, please specify which was used, e.g. 'running crossover'."
  Answer:
  *Included: running*

- **l 133:** "you may want to add that 120 nm was the commonly used distance in CARINA; PACIFICA, and GLODAPv2 so readers understand where this number comes from."
  Answer:
  *Included: which is the distance commonly used in CARINA, PACIFICA and GLO-DAPv2 data products.*

- **l 136:** "the standard deviation .. of what, please specify."

Answer:
*Included: of the offset between two cruises.*

- **Fig 3:** "please specify what the various vertical lines indicate, in the caption."
  Answer:
  *inserted: Crossovers between the cruises 06MT20030723 (blue dots and lines) and the core cruise 33MW19930704-1 (red crosses and lines). The C13 plots show the data and mean profiles of each cruise and the difference plots show the difference profiles with its standard deviation (black lines) as well as the crossovers offset with its standard deviation (red lines).*

- **I 169:** "I do not completely understand, according to Table 4 in the supplement the crossover difference between the 06MT19941012 and the 33MW1993 cruise is not significant, still 0.07 permil is -indirectly-stated in the text. Please also clarify what it takes for a crossover to be significant."
  Answer:
  A significant crossover is a crossover that is based on enough samples to apply the statistics. Most crossovers of this cruise were with the other Meteor cruises, which had to be adjusted even more than this cruise. By applying an adjustment larger than 0.07 permil, the non-significant crossover with cruise 33MW1993 lead to the conclusion that the bias of cruise 06MT19941012 is then overcompensated.
  *The sentence was changed to:*
  *The MLR analysis revealed a smaller offset of $0.05$ and, thus, the cruise was adjusted by -0.07 .*

- **II 219-224:** "This passage is a bit confusing, please clarify. As I understand it, the 2002 Thalassa cruise data were not adjusted, but it had crossovers, why doesn't these data show up in Fig. 5?"
  Answer:

The first Thalassa cruise had only a few samples within the crossover radius, but not enough to be a significant crossover with statistics and a reliable offset. The comparison of both Thalassa cruises suggest, that also the first Thalassa cruise needs to be corrected. Since the samples from both cruises were analyzed in the same lab, it might be reasonable to correct both cruises with the same offset. The MLR revealed now an offset of the cruise 35TH20020611 that is in the same order as the correction suggested for cruise 35TH20060521 by the crossover routine. Unfortunately, the cruise 35TH20060521 could not be compared via the MLR analysis since we did not have DIC data. We now decided to correct both cruises.

*'just a few' in line 21 replace with: only*

*The following sentences were added at the end of the paragraph:*

*The MLR analysis reveal an offset of the 35TH20020611 cruise of -0.23, which is in the same order as the correction suggested by the crossover routine for cruise 35TH20060521. Since the MLR offset for cruise 35TH20020611 is based only on five samples, we applied an adjustment of -0.25 to secure the internal consistency of these two cruises.*

- **l 227:** "this is not correct; the 58GS2003 cruise can be used and is available at CDIAC."

  Answer:

  Yes, that's true, other cruises exist in that area. But, as said above, with the used crossover criterion of 3x3° no crossover with the cruises 58GS20030922 and 74JC20120601 could be observed. The MLR analysis based on the core cruise did not reveal reliable offsets for the cruises that reached into the Nordic Seas.

**Supplement:**

Manuscript prepared for Earth Syst. Sci. Data
with version 2015/04/24 7.83 Copernicus papers of the LATEX class copernicus.cls.
Date: 15 July 2016

[revised manuscript text omitted]

[1] Leibniz Laboratory for Radiometric Dating and Isotope Research, Kiel, Germany

**2   Data Provenance and Structure**

This dataset comprises data and metadata from 32 research cruises/campaigns from several international research groups, in total 6820 samples. Some of these consist of multiple cruises and one is a time series. For the crossover analysis, some consecutive cruises whose data were analyzed together were treated as one cruise. While the focus is on the North Atlantic, four cruises were included that also have stations in the Nordic Seas, and one cruise extends into the South Atlantic. Thereby, consistency with future extended quality controlled datasets for these regions is ensured. Since only deep (> 1500 m) samples of each cruise are compared in this study, only cruises with at least one deep station could be included in this analysis.

Figure 1 shows the locations of all stations with $\delta^{13}$C-DIC data that are part of this compilation. Table 1 shows a summary of the respective cruise dates, the responsible PI and publications in which the data was used. 
[revised manuscript text omitted]

Another method for revealing systematic deviations between different cruises is a regional multi-linear regression (MLR) (Wanninkhof et al., 2003; Jutterström et al., 2010). In this work, a MLR

[Figure]

**Figure 3.** Crossovers between the cruises 06MT20030723 (blue dots and lines) and the core cruise 33MW19930704-1 (red crosses and lines). The C13 plots show the data and mean profiles of each cruise and the difference plots show the difference profiles with its standard deviation (black lines) as well as the crossovers offset with its standard deviation (red lines). The left hand plot shows the original and the right hand plot the adjusted data. In both cases the distribution of the $\delta^{13}$C-DIC on equal density surfaces (left hand side) as well as the mean offset between both cruises (right hand side) is shown. The cruise 06MT20030723 was adjusted by -0.15‰.

based on core cruise data (deeper than 1500 m) was used to verify the suggested corrections, which resulted from the crossover analysis. Moreover, some cruises without a statistically evaluable crossover could now be related to the other cruises. The following equation was used,

$$\delta^{13}\text{C-DIC}_{\text{MLR}} = -16.9 + 0.80 \cdot \text{S} - 0.080 \cdot \Theta - 0.0045 \cdot \text{DIC} \tag{3}$$

with $\delta^{13}$C-DIC$_{\text{MLR}}$ being the calculated $\delta^{13}$C-DIC, S the salinity, $\Theta$ the potential temperature in $^\circ$C and DIC the DIC concentration in $\mu$molkg$^{-1}$. The DIC concentration was chosen because it is strongly related to changes in the isotope composition and DIC data were available for most cruises. Adding more parameters to the MLR, such as apparent oxygen utilization (AOU) or nutrient concentrations, did not improve the agreement between $\delta^{13}$C-DIC and $\delta^{13}$C-DIC$_{\text{MLR}}$ of the core cruise and reduced the amount of cruises hat could be compared via the MLR analysis. The limitation of this method is, of course, that the further away in space and time the cruises are from the core cruise, the more likely an observed offset is real. Especially, the cruises reaching into the Nordic seas

show significant deviations, which are most likely real differences between the basins. Therefore, the offsets revealed by the MLR analysis were not taken into account for these cruises.

**4    Adjustments**

175    The data of all cruises as well as locations are shown in Figure 4. The offsets, as well as the corrections suggested by the WDLSQ inversion routine, the MLR analysis, and the final adjustments are listed in Table 4. In Figure 5 the results of the WDLSQ inversion are shown before and after the adjustments were applied. Some cruises show quite big deviations from the core cruise. However, we do not know the reason for these biases. Besides the actual sample analysis in the laboratory,

180    also different sampling routines on board the ship, insufficient poisoning and the sample storage time can cause these biases. For most cruises that took place in the North Atlantic, the offsets revealed by the MLR analysis were in the same order and magnitude as the suggested correction by the crossover inversion routine. Cruises reaching far into the Nordic Seas or the South Atlantic show huge differences, which are caused by different water mass properties in these areas.

185    A detailed overview of the offset of each crossover in the original as well as the adjusted dataset is given in Table 5 in the supplementary information. Moreover, the evidence for our decision will be presented for each cruise.

**4.1    06MT19941012, cruise #1**

This cruise on the German R/V Meteor is also known as M30-2. The inversion suggested a correction

190    of -0.07 ‰. The mean offset of all crossovers is 0.11 ‰ too high. The MLR analysis revealed a smaller offset of 0.05‰ and, thus, the cruise was adjusted by -0.07 ‰.

**4.2    06MT19970515, 06MT19970707 and 6MT19970815, here referred to as 06MT1997-M39, cruise #2**

These cruises are also known as M39 cruises with three legs of $\delta^{13}$C-DIC sampling (M39-2, M39-4,

195    M39-5). Since each leg of this cruise had only a few stations with $\delta^{13}$C-DIC samples, and all these samples were analyzed together, these cruises were summarized for the crossover study. Both, the inversion routine and the single crossover with the adjusted cruises show no evidence for an offset.

**4.3    06MT19990711 and 06MT19990813, here referred to as 06MT1999-M45, cruise #3**

These cruises are also known as M45-2 and M45-3. Since both were analyzed together, they were

200    summarized for this crossover study. The inversion suggested a correction of -0.15 ‰ and the mean offset of all crossovers was 0.16 ‰ too high. After applying this adjustment and comparing this cruise to the adjusted dataset, the inversion routine still suggested a small correction. Therefore, an adjustment of -0.20 ‰ was applied.

**Table 4.** Overview of all cruises in this dataset. The data of some cruises were combined for the analysis. For more information, please see the detailed description in the 'Adjustments' section. The mean offsets of the crossovers and the MLR as well as the corrections suggested by the WDLSQ inversion for the original and the adjusted dataset are shown. In the last column the applied adjustments are displayed. NC indicates that these cruises were not considered in the inversion since they had no statistically significant crossover and the core cruise is marked with C. Cruises with insufficient quality data are denoted 'poor' and not included in the further analysis.Cruises marked with a * had less than 10 deep samples that were part of the MLR analysis.

| cruise ID | Expocode | Calculated offset | | Suggested correction | | Final adjustments |
| --- | --- | --- | --- | --- | --- | --- |
| | | crossover | MLR | WDLSQ | WDLSQ (adj) | |
| | | / ‰ | / ‰ | / ‰ | / ‰ | / ‰ |
| 1 | 06MT19941012 | 0.11 | 0.05($\pm$0.07) | -0.07($\pm$0.10) | -0.01($\pm$0.02) | -0.07 |
| 2 | 06MT1997-M39 | -0.02 | -0.01($\pm$0.05) | 0.01($\pm$0.14) | 0.00($\pm$0.01) | 0 |
| 3 | 06MT1999-M45 | 0.16 | 0.15($\pm$0.07) | -0.14($\pm$0.09) | 0.00($\pm$0.01) | -0.20 |
| 4 | 06MT20010507 | 0.16 | 0.30($\pm$0.09) | -0.24($\pm$0.10) | 0.00($\pm$0.01) | -0.30 |
| 5 | 06MT20030723 | 0.14 | 0.15($\pm$0.08) | -0.15($\pm$0.09) | 0.00($\pm$0.01) | -0.15 |
| 6 | 06MT20040311 | -0.14 | -0.13($\pm$0.09) | 0.10($\pm$0.09) | 0.01($\pm$0.01) | 0.10 |
| 7 | 316N19970717 | 0.17 | 0.03($\pm$0.05) | -0.06($\pm$0.17) | -0.01($\pm$0.01) | -0.05 |
| 8 | 316N19970815 | | -0.01($\pm$0.05) | | | NC |
| 9 | 316N20030922 | | 0.11($\pm$0.02)* | | | NC |
| 10 | 316N20031023 | | | | | NC |
| 11 | 33RO19980123 | | -0.03($\pm$0.04) | | | NC |
| 12 | 33MW19910711 | -0.02 | -0.04($\pm$0.11) | 0.00($\pm$0.01) | 0.00($\pm$0.01) | 0 |
| 13 | 33MW19930704 1 | -0.05 | 0.00($\pm$0.04) | 0.00($\pm$0.01) | 0.00($\pm$0.01) | C |
| 14 | 35TH20020611 | | -0.24($\pm$0.07)* | | | 0.25 |
| 15 | 35TH20060521 | -0.39 | -0.02 | 0.24($\pm$0.21) | -0.03($\pm$0.05) | 0.25 |
| 16 | 58GS20030922 | | -0.15($\pm$0.07) | | | NC |
| 17 | 58JH19920712 | | -0.11($\pm$0.10) | | | NC |
| 18 | 58JH19940723 | | -0.06($\pm$0.05)* | | | NC |
| 19 | 64TR19900417 | | | | | poor |
| 20 | 74DI20120731 | -0.33 | -0.14($\pm$0.13) | 0.13($\pm$0.28) | 0.12($\pm$0.12) | 0 |
| 21 | 74JC20120601 | | -0.33($\pm$0.11) | | | NC |
| 22 | 74DI20140606 | -0.11 | -0.09($\pm$0.09) | 0.07 | | 0 |
| 23 | OMEX1NA | -0.14 | -0.23($\pm$0.15) | 0.03($\pm$0.13) | 0.02($\pm$0.02) | 0 |
| 24 | 316N19810401 | -0.06 | -0.04($\pm$0.08) | -0.03($\pm$0.10) | -0.01($\pm$0.03) | 0 |

[Figure]

**Figure 4.** Adjusted $\delta^{13}$C-DIC profiles and locations of each cruise. The green profiles represent the data of the specific cruise whereas the gray dots show all profiles in the dataset. *continued on next page*

[Figure]

**Figure 4.** *continued from previous page.*

Adjusted $\delta^{13}$C-DIC profiles and locations of each cruise. The green profiles represent the data of the specific cruise whereas the gray dots show all profiles in the dataset.

**4.4 06MT20010507, cruise #4**

205 This cruise is also known as M50-1. The inversion routine suggested a correction of -0.24‰, whereas the mean offset was 0.16 ‰ too high. The MLR analysis revealed an offset of 0.30 ‰. Based on the southern crossover with cruise 06MT20040311 and 316N19970717 an adjustment of -0.30 ‰ was applied.

**4.5 06MT20030723, cruise #5**

210 This cruise is also known as M59-2 (Friis et al., 2007). The correction suggested by the inversion routine is -0.15‰, which matches with the positive offsets of the crossovers, except of those with 33TH20060521. Based on the crossover with the core cruise, an adjustment of -0.15 ‰ was applied.

[Figure]

**Figure 5.** The results of the WDLSQ based inversion routine for the original (blue circles) and the adjusted dataset (red stars). The cruises are plotted at the time the data were collected vs. the suggested correction.

**4.6   06MT20040311, cruise #6**

This cruise is also known as M60-5. The inversion routine indicates that the $\delta^{13}$C-DIC data of this cruise are 0.10 ‰ too low. Additionally, the mean offset as well as the MLR analysis shows that these data are too low. An adjustment of +0.10 ‰ was applied.

**4.7   316N19970717, cruise #7 and 316N19970815, cruise #8**

These cruises followed the WOCE/GO-Ship standard lines A20 and A22. The inversion suggests a correction of -0.06 ‰ for 316N19970717. It shows one crossover with cruise 06MT2004031, in which a significant positive offset is still visible after cruise 06MT20040311 was corrected. Therefore, an adjustment of -0.05 ‰ was applied for cruise 316N19970717. The cruise 316N19970815 does not show a statistically significant crossover.

**4.8   316N20030922, cruise #9, and 316N20031023, cruise #10**

These cruises, which took place in the tropical western Atlantic, following the A20 and A22 lines, have only one deep station each. The crossovers of these stations with both, the adjusted data of cruise 06MT20040311 and cruise 316N19970717 show a good agreement, suggesting that no adjustment should be applied.

**4.9   33RO19980123, cruise #11**

This cruise has one statistically insignificant crossover with the cruise 06MT20040311 and one with cruise 33MW19930704-1. Both seem to be in good agreement, suggesting that no adjustment should be applied.

**4.10  33MW19910711, cruise #12, and 33MW19930704-1, cruise #13**

The cruise 33MW19930704-1 was considered as core cruise in the present analysis. The cruise 33MW19910711 extents into the south Atlantic and its crossover with cruise 13 shows no need for an adjustment.

**4.11  35TH20020611, cruise #14, and 35TH20060521, cruise #15**

The latter of these two cruises has a few quantitative crossovers, which show a high offset of -0.39‰. Furthermore, the inversion suggests a correction of 0.24‰. The high variability of the sampling area south of Iceland, as well as an increasing lightning of the deep water carbon pool over time don't give an adequate explanation for this large deviation and, therefore, an adjustment of -0.25‰ was applied. The cruise 35TH20020611 shows only qualitatively analyzable crossovers, which show a lighter carbon pool compared to earlier cruises and a heavier one compared to the original data of cruise 35TH20060521 (Racapé et al., 2013). After adjusting cruise 35TH20060521, both cruises, which were analyzed in the same laboratory, are not in good agreement anymore, which suggests that the earlier cruise also has too low isotope values. The MLR analysis reveal an offset of the 35TH20020611 cruise of -0.23‰, which is in the same order as the correction suggested by the crossover routine for cruise 35TH20060521. Since the MLR offset for cruise 35TH20020611 is based only on five samples, we applied an adjustment of -0.25‰ to secure the internal consistency of these two cruises.

**4.12  58GS20030922, cruise #16**

This cruise has only two very weak crossover, one with the TTO data, which took place 30 years earlier and one with 74JC20120601. When comparing the 58GS20030922 to the latter the offset seems to be consistent with an increasing lightning of the DIC caused by an increasing amount of anthropogenic carbon, that is decreasing with increasing depth. However, the crossover with the TTO data is not consistent with this. Therefore, no adjustment was applied.

**4.13  58JH19920712, cruise #17, and 58JH19940723, cruise #18**

These two cruises took place in a highly variable area.No statistically relevant crossover exists but the data are in good agreement with the core cruise and the other adjusted cruises in that area.

**4.14  64TR19900417, cruise #19**

This cruise shows extreme scatter compared to all other cruises and, therefore, was not included into the adjusted product. When comparing crossover stations this cruise shows a mean offset to other cruises of about -1.2‰.

**4.15 74DI20120731, cruise #20**

Both the inversion and the offset mean of the crossover suggest a correction of +0.13 ‰ for the cruise (Humphreys et al., 2015). This most recent cruise took place near the Scotland-Iceland ridge where the deep water masses cannot be assumed to be constant over time. All crossovers indicate a lower $\delta^{13}$C-DIC of this cruise when comparing it with the others, which is consistent with an increased amount of anthropogenic carbon. Therefore, no adjustment was applied.

**4.16 74JC20120601, cruise #21**

This cruise has only a few stations with $\delta^{13}$C-DIC data in a highly variable region. It has only one crossover with the cruise 58GS20030922. In the MLR analysis, this cruise is too far away from the core cruise to give a reliable outcome. No adjustment was applied.

**4.17 74JC20140606, cruise #22**

This cruise covers the North Atlantic between Canada, Greenland and Scotland. The crossover inversion gives a suggested correction of 0.07 ‰ and the MLR analysis an offset in the same magnitude -0.09 ‰. Since this cruise took place 20 years after the core cruise, anthropogenic influences cannot be neglected in this case. Therefore, no adjustment was applied.

**4.18 OMEX1NA, cruise #23**

During the OMEX1 project in the North Atlantic $\delta^{13}$C-DIC samples were taken in January 1994. The MLR analysis revealed an offset of -0.26 ‰. In contrast to that, the crossover inversion did not suggest a correction. No adjustment was applied.

**4.19 316N19810401, cruise #24**

The cruises 316N19810401, 316N19810416, 316N19810516, 316N19810619, 316N19810721, 316N19810821 and 316N19810923 are combined and usually named Transient Tracers in the Oceans North Atlantic Study (TTO-NAS). The inversion does not suggest any correction for this dataset.

**5 Conclusions**

The finalized, quality controlled dataset of $\delta^{13}$C-DIC presented here consists of 24 cruises (some of which consists of multiple legs that were grouped) that have been quantitatively compared to each other and form an internally consistent dataset. Nine cruises could not be quantitatively compared to the other cruises due to a lack of crossovers and / or deep $\delta^{13}$C-DIC data. The reason of the

| # | | 06MT19941012 | 06MT1997-M39 | 06MT1999-M45 | 06MT20010507 | 06MT20030723 | 06MT20040311 | 316N19970717 | 316N19970815 | 316N20030922 | 316N20031023 | 33RO19980123 | 33MW19910711 | 33MW19930704-1 | 35TH20020611 | 35TH20060521 | 58GS20030922 | 58JH19920712 | 58JH19940423 | 74DI20120731 | 74JC20120601 | 74JC20140606 | OMEXINA | 316N19810401 |
|---|---|---|---|---|---|---|---|---|---|---|---|---|---|---|---|---|---|---|---|---|---|---|---|---|
| 1 | 06MT19941012 | ● | > | 0.16 (±0.07) | 0.31 (±0.06) | 0.06 (±0.06) | > | = | | | | | | = | | | | | | | | < | -0.10 (±0.03) | < |
| 2 | 06MT1997-M39 | > | ● | 0.21 (±0.07) | < | 0.15 (±0.06) | | | | | | | | = | ^ | -0.32 (±0.06) | | ^ | | | | = | = | = |
| 3 | 06MT1999-M45 | -0.03 (±0.07) | -0.01 (±0.07) | ● | 0.11 (±0.07) | 0.00 (±0.06) | > | < | | ^ | | | | -0.20 (±0.07) | ^ | -0.38 (±0.13) | | < | | -0.31 (±0.15) | | -0.23 (±0.05) | ^ | -0.09 (±0.06) |
| 4 | 06MT20010507 | -0.09 (±0.06) | = | 0.01 (±0.07) | ● | -0.08 (±0.06) | > | ^ | | | | | | | | | | | | | | ^ | ^ | |
| 5 | 06MT20030723 | 0.02 (±0.06) | 0.00 (±0.06) | -0.04 (±0.06) | -0.07 (±0.06) | ● | -0.00 (±0.02) | | | | | ^ | | 0.00 (±0.05) | | -0.48 (±0.09) | | | | ^ | | -0.30 (±0.08) | -0.17 (±0.02) | -0.20 (±0.04) |
| 6 | 06MT20040311 | = | = | = | = | 0.15 (±0.02) | ● | 0.11 (±0.02) | | | | | | = | | | | | | | | | | 0.13 (±0.08) |
| 7 | 316N19970717 | = | | < | < | ^ | -0.00 (±0.02) | ● | < | = | | ^ | | | | | | | | | | | | = |
| 8 | 316N19970815 | | | | | | | ^ | ● | | | | | | | | | | | | | | | |
| 9 | 316N20030922 | | | ^ | ^ | | ^ | = | | ● | | | | | | | | | | | | | | ^ |
| 10 | 316N20031023 | | | | | | | | | | ● | | | | | | | | | | | | | |
| 11 | 33RO19980123 | | | | | ^ | > | | | | | ● | | ^ | | | | | | | | | | = |
| 12 | 33MW19910711 | | | | | | | | | | | | ● | -0.02 (±0.05) | | | | | | | | | | |
| 13 | 33MW19930704-1 | = | = | | = | 0.00 (±0.05) | -0.01 (±0.08) | | | | | ^ | 0.02 (±0.05) | ● | = | | | = | | < | < | < | | ^ |
| 14 | 35TH20020611 | | ^ | | | = | | | | | | | | = | ● | < | | | | ^ | | = | | = |
| 15 | 35TH20060521 | | | 0.18 (±0.13) | | 0.33 (±0.09) | | | | | | | | | ^ | ● | < | | | | | 0.20 (±0.07) | | < |
| 16 | 58GS20030922 | | | | | | | | | | | | | | | ^ | ● | | | | < | | | |
| 17 | 58JH19920712 | | = | < | = | = | | | | | | | | = | | | | ● | = | | | -0.20 (±0.07) | | < |
| 18 | 58JH19940423 | | = | | = | = | | | | | | | | = | | | | = | ● | | | | | |
| 20 | 74DI20120731 | | | | < | < | | | | < | | | | < | ^ | | | | | ● | ^ | < | | < |
| 21 | 74JC20120601 | | | | | 0.15 (±0.08) | | | | | | | | | | | < | | | ^ | ● | | | ^ |
| 22 | 74JC20140606 | < | = | = | ^ | 0.02 (±0.02) | | | | | | | | = | = | | | | | ^ | < | ● | < | ^ |
| 23 | OMEXINA | 0.03 (±0.03) | = | = | | 0.05 (±0.04) | -0.03 (±0.04) | | | | | | | | | | | | | | | | ● | |
| 24 | 316N19810401 | > | = | ^ | ^ | | | < | | < | | = | | ^ | = | ^ | | | | | | ^ | | ● |

**Table 5.** This table shows an overview of all crossovers. The symbol ● in each row divides the table into a triangle in the upper right and one in the lower left of the table. In the upper right corner for each crossover the offsets of the offsets of the original dataset are listed. In the lower left corner the remaining offsets in the adjusted dataset are shown. Not statistically relevant crossovers are displayed with >, < and = indicating the tendency of not significant crossover. Please note that the offsets shown in the table result from $\delta^{13}C_{column} - \delta^{13}C_{row}$. All offsets are given in ‰.

deviations between single cruises could not be revealed. There was no correlation between a cruises' bias or its scatter and storage time, analyzing period or volume of $HgCl_2$ added.

The internal consistency of the adjusted dataset was calculated to be 0.017‰ based on Equation 2.

The database is available at CDIAC via http://cdiac.ornl.gov/oceans/ndp_096/NAC13v1.html, doi:10.3334/CDIAC/OTG.NAC13v1.

**6  Acknowledgment**

We would like to thank all the people, both researchers as well as captains and crews, who spend time at sea and in the lab collecting and measuring the samples and preparing the data, which are presented here, and the PIs for sharing it. The measurements of the majority of previously not public $\delta^{13}C$ data was supported by the Deutsche Forschungsgemeinschaft (DFG) through SFB460 and this work was funded by the Future Ocean Excellence Cluster Project CP1140. M.P. Humphreys is funded by the Natural Environment Research Council (UK) through CaNDyFloSS: Carbon and Nutrient Dynamics and Fluxes over Shelf Systems (NE/K00185X/1). We also thank Aley Kozyr from CDIAC for preparing the NAC13v1-website on CDIAC.

---

## Author Comment (AC2) · 15 Jul 2016

**Response to Reviewer 2**

Thank you very much for this review. We hope that we could address the raised issues in a sufficient way.

Generally, we agree that this dataset for the North Atlantic ocean can only be a start and should, in the best case, be extended to the globe. However, this is limited by the spatial overlapping of the respective cruises. The new $\delta^{13}$C-DIC data of the 6 Meteor cruises, now clearly pointed out to us the need for an internal quality control and at the same time revealed enough crossovers for a reliable analysis.

We did not perform AOU vs. $\delta^{13}$C-DIC plots for identifying outliers in the beginning.

[Figure]

For this review, however, we did so but there were no remaining outliers. (The cruise 64TR19900417 is completely out of the normal range, but it was flagged as bad anyway).

Please note, that changes we did in the manuscript or the dataset are italicized in the following and highlighted in red in the manuscript.

- **p 3:** "Anthropogenic $\delta^{13}$C-DIC changes have been estimated by models (e.g. our 2013 paper mentioned above). These model results could potentially be used to estimate the effect in different regions."
  Answer:
  We agree that these model outputs could be used to estimate the effect of anthropogenic carbon on deep water masses. We decided, however, not to include this analysis in this publication as it's primary purpose is to provide a quality-controlled dataset for all kinds of further analysis.

- **p 7:** "Why does cruise 33MW199930704-1 have high quality data? Were there objective criteria used to determine this?"
  Answer:
  *Sentence changed to:*
  *The cruise 33MW199930704-1 was analyzed by a reputable laboratory, has relatively low scatter and covers wide distances.*
  Of course, it can never be completely excluded, that this data has a bias itself. However, by testing all other cruises against this one, we achieve internal consistency of the presented dataset.

- **l 140:** "these cruises are 10 years apart and I could imagine that at high latitudes anthropogenic $\delta^{13}$C-DIC could have an impact. (see comments above)."
  Answer:
  Yes, both cruises are 10 years apart, but we suggest that 10-year-changes due

to anthropogenic carbon in the deep North Atlantic Ocean (here the region between 40-55° N) are smaller than the measurement uncertainty. Moreover, there is no trend observed with depth as it was found for some crossovers further north where the influence of anthropogenic carbon definitely has to be taken into account. Finally, for this specific crossover, the latter cruise had higher $\delta^{13}$C-DIC data than the earlier one, which is not consistent with what we would expect for an increased amount of anthropogenic carbon in the latter cruise data.

- **Fig 3:** "the font and figure is too small. It is not readable. Please increase size."
  Answer:
  These two pictures are shown only as an example for a typical crossover. *However, we increased the font size.*

- **I 182:** ""-0.20 permil" but in Tab. 3 -0.15 permil is listed. Please check this inconsistency."
  Answer:
  *Inconsistency was corrected.*